SocioPedia+: a visual analytics system for social knowledge graph-based event exploration

Nguyen Tra My 1
Chun Hong-Woo 2
Hwang Myunggwon 3
Kwon Lee-Nam 2
Lee Jae-Min 2
Park Kanghee 2
Jung Jason J. j2jung@gmail.com 1
1 Department of Computer Engineering, Chung-Ang University , Seoul , Korea
2 Korea Institute of Science and Technology Information , Seoul , Korea
3 Korea Institute of Science and Technology Information , Daejeon , Korea
Chen Abel C.H.
Electronic publication date: 2023 Mar 20
Publication date: 2023
Volume: 9
Electronic Location ID: e1277
Received 2022 Nov 3; Accepted 2023 Feb 15
Copyright: ©2023 Nguyen et al.
Copyright year: 2023
Copyright holder: Nguyen et al.
License: This is an open access article distributed under the terms of the Creative Commons Attribution License, which permits unrestricted use, distribution, reproduction and adaptation in any medium and for any purpose provided that it is properly attributed. For attribution, the original author(s), title, publication source (PeerJ Computer Science) and either DOI or URL of the article must be cited.
License URL: https://creativecommons.org/licenses/by/4.0/

Keywords: SocioPedia, Knowledge graph, Event detection and exploration, Visual analytics

Funding: The Chung-Ang University Young Scientist Scholarship in 2021 Ministry of Science and ICT(MSIT) grant by the Korean government K-22-L03-C02 This research was supported by the Chung-Ang University Young Scientist Scholarship in 2021. This work was also supported by the Ministry of Science and ICT(MSIT) grant by the Korean government. (KISTI Project No. K-22-L03-C02). The funders had no role in study design, data collection and analysis, decision to publish, or preparation of the manuscript.

==============================
In the recent era of information explosion, exploring event from social networks has recently been a crucial task for many applications. To derive valuable comprehensive and thorough insights on social events, visual analytics (VA) system have been broadly used as a promising solution. However, due to the enormous social data volume with highly diversity and complexity, the number of event exploration tasks which can be enabled in a conventional real-time visual analytics systems has been limited. In this article, we introduce SocioPedia+, a real-time visual analytics system for social event exploration in time and space domains. By introducing the dimension of social knowledge graph analysis into the system multivariate analysis, the process of event explorations in SocioPedia+ can be significantly enhanced and thus enabling system capability on performing full required tasks of visual analytics and social event explorations. Furthermore, SocioPedia+ has been optimized for visualizing event analysis on different levels from macroscopic (events level) to microscopic (knowledge level). The system is then implemented and investigated with a detailed case study for evaluating its usefulness and visualization effectiveness for the application of event explorations.

Introduction

Recent decades have witnessed the extensive development of social networks, which are the most convenient platforms for exploring up-to-date information. Owing to the enormous volume of collected social data with high diversity and fast updated, an extensive spectrum of ongoing topics and events across the globe can be robustly covered and actively shared in social media platforms. As a consequence, the social network has become a valuable knowledge source which attracts a lot of attention for applying in a vast of life domains, such as crisis management, politics analysis, and health-care systems (Adiyoso, 2022; Tran et al., 2022; Pomare et al., 2019). To derive valuable insight with multivariate analysis, considerable efforts have been conducted to the research on visual analytics (VA) systems for spatiotemporal event exploration in recent years to enhance human information analysis processes and provide intuitive approaches for promptly seeking, determining, and further exploring the hidden patterns of social events in both time and space domains. VA systems can offer excellent opportunities to understand what is the initial factor creating an event, its evolutional patterns following time and space, its causality, and even enhance the prediction for future forecast. Furthermore, text content analysis and sentiment analysis of VA systems would provide different aspects to investigate response and behaviors of different users or communities. To the date, a vast of exploration tasks have been reported and constructed in several VA systems. The main tasks can be categorized into event identification, evolution analysis, comparison analysis, diffusing, and causality analysis. Nevertheless, it should be noted that although there are diverse of reported VA systems focusing on individual tasks of spatiotemporal event exploration, the recent literature of research on VA systems has been lacked of VA systems which can perform full event exploration processes. Increasing the number of aspects for multivariate analysis is a useful method to enable more tasks for both visual analysis and event explorations. However, there exists a limitation on multivariate analysis of reported VA systems, which is typically hard to exceed two or three analysis aspects.

To observe the characteristics in both temporal and spatial dimensions of social information, reported VA systems focus on collecting and investigating social data based on user generated contents (text contents, uploaded multimedia files, emoji, hashtag), interests, activities, and connection networks. However, there is a large amount of collected knowledge from social networks having unstructured short-text forms since only bag of words are typically extracted and analyzed in the collecting and analyzing processes of VA systems. One of the potential way of extracting and processing efficiently the for visualizing the massive knowledge from social networks is a knowledge graph (KG) based on visualizing information in the form of facts, including entities and the connections between them (Chen et al., 2020). All entities and their related relationships, are stored in the form of triples {subject, predicate, object}, where the subject and object are the head entity and tail entity, the predicate is a relation between them. The formal, semantic, and organized depiction of information as a network of interconnected nodes and edges is known as knowledge graphing. This allows VA systems to capture and process knowledge more efficiently and unambiguously. However, most reported works collected the information from social networks only as static facts, which means that facts are not changed over time. Therefore, social knowledge has typically been collected and visualized without associated with its temporal factors, which would drive to the lack of information or incorrect information. For instances, the fact {}Barack Obama, president of, United States {} is only valid from 2009 to 2017; thus if VA systems can only extract facts without temporal information, the collected information for event exploration might be inaccurate information. One of the main difficulties of mining temporal information from social data is caused majority of data do not include time information. Due to that limitation, VA systems typically use static facts and only visualize them based on the time period of facts-appearance on the social stream, which might cause to inaccurate information for event exploration processes (Cashman et al., 2020).

In this article, we introduce SocioPedia+ (https://github.com/kecau/sociopedia), a visual analytics system for spatiotemporal event exploration using social knowledge graph. By introducing one more dimension of social knowledge graph into the multivariate analysis, the process of event explorations can be efficiently enhanced owing to the enabling of full required tasks for visual analytics and event explorations. Our key contributions of the article are as follows.

• We propose a method to efficiently extract temporal information from a social network. Different with conventional techniques which are strongly based on finding the temporal words from the raw text, we propose to use knowledge occurrence and diffuse-degree to identify and extract the time-valid period of each extracted knowledge.

• We propose different visualization techniques for efficiently exploring social knowledge and social event in an intuitive way which considers both temporal and spatial aspects. The visualization is optimized in two levels for representing social knowledge and social events with multivariate analysis. To provide an insightful view, social knowledge is represented in three visualizations, including static visualization (for providing initial overview), timeline visualization (for analyzing knowledge features over time, identifying relationship of knowledge in a specific time period, and analyzing knowledge evolution), and dynamic timeline visualization (for intuitively observing and monitoring the variation of knowledge and their attributes over time). Meanwhile, we propose a novel social-events visualization to efficiently perform their sentiment analysis following different countries and promptly exploring the social events.

• We present an end-to-end system for spatiotemporal social knowledge visual analytics performing most of visual analytics main tasks. On the perspective view for social knowledge analysis, SocioPedia+ visualizes knowledge in an easy-to-understand, intuitive way to identify old/new, important/unimportant, the important time-period in which the knowledge was mentioned, the connectivity, and evolution of different knowledge. On the other hand, social event exploration process can receive considerably advantages from SocioPedia+ to promptly have an insightful view for the events, such as what is the social knowledge from event, what is the social events, whether the response from different countries are negative or positive and what is the reason, and so on.

The rest of this article is organized as follows. The “Related Work” section introduces the background knowledge of related studies. Next, “Event Exploration Requirements” describes general requirements for event exploration and visualization, which has been highlighted via a real-world survey. In “SocioPedia+ Architecture”, data presentation and detail of our module are presented while “Visualization Techniques” provides an insight view on proposed visualization techniques. The experimental analysis for the proposed VA system is later mentioned in “Evaluation” section. Finally, “Conclusions and Future Work” concludes this study and discusses about potential future works.

Related Work

The most related research to SocioPedia+ is VA systems for spatiotemporally exploring social events in real-time, which will be discussed in “Visual analytics system for social event explorations”. Meanwhile, the sections “Temporal information extraction” and “Visualizing knowledge over time” will list the recent studies in temporal information extraction and techniques for effective knowledge visualization over time and space as well as highlight the disadvantage of recent techniques which makes them difficult to apply in a real-time VA system.

Temporal factor extraction

Due to the enormous volume with unstructured data collected from a social networks, extracting knowledge with their temporal information is quite challenging, thus recently attracts a considerable attention. To the date, reported works on extracting temporal information can be divided into two major groups of deep-learning based (DL-based) method and temporal knowledge harvesting (TKH) method. Coupling system and extracting system are two main directions of TKH for extracting the temporal information. While coupling system uses the relation facts as the inputs and determines key sentences to extract relevant temporal information (Cucerzan & Sil, 2013; Garrido, Penas & Cabaleiro, 2013; Talukdar, Wijaya & Mitchell, 2012; Wang et al., 2019), TKH extracting systems are able to deriving temporal information from scratches by analyzing the textual pattern of each relation from raw text. In the comparison, TKH coupling system cannot simultaneously detect both the relations and their respective temporal information due to the lack of ability for directly deriving temporal factor from raw text, whereas TKH extracting systems able to simultaneously detect all relations including temporal factors (Wang et al., 2010; Wang et al., 2011). Some good examples for extracting TKH system are T-Yago (Wang et al., 2010) and the Pravda system (Wang et al., 2011). While the former one extracted the valid time information by collecting semi-structure data from Wikipedia’s Infoboxes, the later system represent collected facts by using their textual pattern. After that, the extracted facts will be labeled with a graph-based label propagation algorithm. Meanwhile, DL-based works on extracting relational facts have recently been dominant than DL-based works for extracting temporal facts (Su et al., 2022; Zhou et al., 2022; Sun et al., 2021). Only few temporal extraction works using DL-based techniques have been reported while most of them have been struggled by several considerable issues. The major issue is coming from the supervised or unsupervised of DNN model, thus requiring them to be trained with a large scale labeled dataset. However, the enormous volume of noisy, unstructured, and ambiguity social data has raised many challenges for the construction of a high accuracy labeled dataset. Furthermore, the requirements of high computing volume of DL-based technology has constrained its applicability for implementing a real-time VA systems. Generally, despite of using different approaches for finding the time information of knowledge, all mentioned techniques require the existence of temporal words in original text. For example, the temporal words can be hour, date, month, or year. If the original text does not include any temporal words, those techniques would not able to extracting temporal information. This is one of the biggest challenge on extracting temporal information from social networks because most of social data have been shared without including any temporal words.

Visualizing knowledge over time

To intuitively represent the entities and their relations, typical KG visualization has utilized nodes and edges system for demonstrating a triples of SPO—{subject, predicate, object}. Recently, a considerable number of works have stated that considering social knowledge as static fact is not adequate since a lot of facts are changed over time. Therefore, reported works have introduced the temporal factor, making the triple of SPO become SPOT with T standing for timestamp or temporal factor. Consequently, a quadruple of SPOT is used to demonstrate the knowledge associated with its temporal information. Most of recent works with temporal knowledge graph (TKG) have put the time start and time end of each knowledge inside brackets and put it near the edge of predicate. Nevertheless, this visualization is only suitable if the data volume is not too big. Besides that, by listing all temporal information of collected knowledge simultaneously in such form, users would hardly to achieve an intuitive analysis perspective due to the enormous volume of collected knowledge and poor capability on comparing temporal facts from different knowledge.

For enhancing the efficiency of temporal knowledge visualization, reported works have introduced different techniques to include the timeline into the visualization. For instance, an event-centric temporal KG visualization following timeline has been reported in EventKG (Gottschalk & Demidova, 2018). To address the issue of only 33% WikiData events containing temporal information, the authors proposed an event-centric visualization which uses the center nodes for representing event names whereas surrounding nodes and edges referring to related entities and relationships. The sequence of event nodes will be put on a unify timeline to provide a more comprehensive overview for users. Nevertheless, this visualization is only suitable for demonstrating event-centric data. Besides that, for applying such visualization for event exploration, the visualization should include more information to reduce the aggregation efforts of users. Meanwhile, WikiTimeline (Graux et al., 2021) has been reported to optimizing the visualization effectiveness for entity-centric KG. The authors has constructed a completely different visualization which have left and right vertical axis designated for presenting the entities and horizontal bottom axis designated for presenting the timeline. The relation between entities are symbolized as a bar chart with the length representing for the time period of that relations. Although the reported visualization is a creative solution, rendering with large datasets like data from social media is difficult and has many limitations.

Visual analytics systems for social event explorations

Table 1 has provided the survey of reported VA systems evaluated under three main categories including VA system main tasks, event exploration tasks, and the capability of multivariate analysis. In particular, important VA processes include six main tasks of virtual monitoring, feature extraction, event detection, anomaly detection, predictive analysis, and situation awareness. Meanwhile, the research on event exploration can focus on different tasks, such as event identification, evolution analysis, comparative capability, information diffusion analysis, and causality analysis. The multivariate analysis capability of each reported VA systems has also included in the survey table to provide a fair comparison.

Table 1 Survey of reported visual analytics system for social event exploration in spatial–temporal domains, evaluated following three categories of visual analytics main tasks, event exploration main tasks, and multivariate analysis capability.

	Whisper (Cao et al.)	D-map+ (Chen et al.)	STempo (Robinson et al.)	Socialwave (Sun et al.)	E-map (Chen et al.)	Scatterblogs2 (Bosch et al.)	Twitinfo (Marcus et al.)	StreamExplorer (Wu et al.)	ScatterBlogs v2.0 (Chae et al.)	TwitterScope (Gansner, Hu & North)	ScatterBlogs v1.0 (Thom et al.)	SocioPedia+	
Main steps of VA	Virtual monitoring	●	●	–	●	●	●	●	●	–	–	●	●	
Feature extraction	–	●	–	–	–	–	–	–	–	●	–	●	
Event detection	●	–	–	–	–	–	●	●	●	–	–	●	
Anomaly detection	–	●	●	–	–	●	–	–	●	–	●	●	
Predictive analysis	–	◐	–	◐	–	–	–	◐	–	–	–	◐	
Situation awareness	–	◐	–	◐	–	–	–	◐	–	–	–	◐	
Event exploration	Event identification	–	●	–	◐	◐	●	●	◐	◐	–	◐	◐	
Evolution analysis	–	●	●	●	●	–	–	●	–	–	–	●	
Comparative capability	–	●	–	●	–	–	–	◐	–	–	–	●	
Diffusing analysis	●	●	●	●	●	–	●	–	–	–	–	●	
Causality analysis	–	◐	–	–	–	–	–	◐	–	–	–	◐	
Multivariate analysis	Event distribution - Spatial	–	●	●	●	–	–	●	●	●	–	●	●	
Event distribution - Temporal	–	–	●	●	–	–	●	●	●	–	●	●	
Keyword analysis - Spatial	–	–	–	●	●	●	–	–	●	–	●	–	
Keyword analysis - Temporal	–	–	–	●	●	●	●	●	●	●	●	●	
Topic analysis - Spatial	–	–	–	–	●	●	–	–	●	–	–	–	
Topic analysis - Temporal	–	–	–	–	●	●	–	●	–	●	●	●	
Knowledge analysis - Spatial	–	–	–	–	–	–	–	–	–	–	–	●	
Knowledge analysis - Temporal	–	–	–	–	–	–	–	–	–	–	–	●	
Sentiment analysis - Spatial	●	–	–	–	–	–	–	–	–	–	–	●	
Sentiment analysis - Temporal	●	–	–	–	–	–	●	●	–	–	–	●	
Notes.

● Features which are directly provided by the reported system

◐ Features which are not directly provided but can extract through multivariate analysis of reported system

– Reported system does not provide this function

Due to the complexity and enormous volume of structure/unstructured, heterogeneity, fast-changed, and hindered data, effectively processing and intuitively visualizing collected knowledge from social networks is quite challenging. To the date, the number of tasks for visual analytics and event exploration in reported VA systems have been limited, as can be observed in Table 1. Particularly, only the works of D-Map+ (Chen et al., 2018) and StreamExplorer (Wu et al., 2017) could cover 4/6 required tasks of a VA system while other reported works are typically able to performing only 2/6 required tasks. Meanwhile, D-Map+ is more complete than other works when event exploration tasks are considered. Among all reported VA system, only D-Map+ VA system can provide full event exploration tasks. On the other hand, VA system needs to provide multivariate analysis which can allow the users to observe, aggregate the information from as many aspects and levels as possible. While analyzing event distribution following time and space dimensions has been reported in most VA system, the characteristic analysis of other social objects (keywords, topics, sentiments) also need to be included. However, it is not easy to extract those characteristic due to the complexity of collected social data. Consequently, number of multivariate analysis which reported VA systems can provide hardly exceed 5/10, as depicted in Table 1. It also should be noted that temporal and spatial sentiment analysis are typically not included in majority of reported VA systems due to the challenge on effectively visualize multiple information which can help the users aggregate data at a glance. Besides that, only few works have considered the sentiment analysis in both temporal and spatial dimensions. As a consequence, the number of multi-perspective analysis in real-time VA system has recently been limited, thus constraint VA system ability on fulfilling full required tasks of event exploration.

In SocioPedia+, we provide a multilevel analysis which are optimized for event analysis (derived from event distribution analysis and sentiment analysis) and knowledge analysis for providing more detail look on the process of event exploration. To provide an insightful view, social knowledge is represented in three visualizations, including static visualization (for providing initial overview), timeline visualization (for analyzing knowledge features over time, identifying relationship of knowledge in a specific time period, and analyzing knowledge evolution), and dynamic timeline visualization (for intuitively observing and monitoring the variation of knowledge and their attributes over time). Both temporal and spatial aspects have been included in all analysis level to provide the most insightful knowledge and events understanding. In addition, it should be highlighted that only SocioPedia+ has provided such knowledge analysis in temporal and spatial domains to further enhance the exploration processes. On the other hand, we propose a novel social-events visualization to efficiently perform their sentiment analysis following different countries while promptly exploring the social events. Owing to broad covering range of multivariate analysis, SocioPedia+ can satisfy all tasks in the visual analytics pipeline and required tasks for event exploration. As a consequence, SocioPedia creates considerably ideal environment for users analyzing collect social data at a glance.

Event exploration requirements

To design SocioPedia+ to efficiently handle the challenge of data analyst on social event explorations in both time and space domains, we have interviewed and discussed with five different candidates from different countries, including a professor, two graduate student researchers, and two business analyst/data analyst working on software companies. All of the participants are not co-authors of this articles while they have background in computer science and frequently work with event exploration from social media networks to support their research and business. The discussion aims to highlight how an event explorer would like to collect, aggregate, and further investigate with the collected data from social networks. In addition, the difficulties and challenges when they are working with other event exploration VA tools have also been surveyed and thoroughly discussed. Based on their feedbacks combined with our literature review on recent reported VA systems for event explorations, we derived several requirements for a VA systems designed for social event explorations. From different perspective views, scope of some requirements can be overlapped. However, to giving the most detailed information, we included all requirements and categorized it, as follows:

V Visualization requirements.

V1 Real-time monitoring capability. The motivation for real-time monitoring is to gain a quick overview and can help the users easily determine several key analysis. To efficiently achieve this, the system should provide an intuitive overview visualization with several key and dynamic updated analysis.

V2 Displaying anomaly time periods for intuitively and automatically detecting events. Because the amount of data collected from social networks is enormous, manually detecting the anomaly patterns is not an easy task for users. On the other hand, if the systems automatically detect events without efficiently displaying them, the users interactive would be considerably limited. Therefore, the VA system must simultaneously satisfy both of them.

V3 Aggregate events at a glance. The VA system should help users easily and promptly to have brief information and analysis about the events, thus users can decide if the events are worth for further analysis or not based on those information.

V4 Multi-perspective analysis. Social data can reflect many valuable information through not only their contents but also on the users behaviors and other perspective view. By providing a multi-perspective point of views, the users can have a more insightful view and further enhance their analysis.

V5 Capable of comparative visualization on different perspectives. Comparative is one of crucial tasks for making the users truly understand about events. Several aspects can be considered to put on the comparison, such as event distribution on time and space domains, sentiment variation following time and space, extracted knowledge from contents. Based on such information and comparisons, users can more efficiently determine the common and different points between events.

V6 Event diffusion monitoring. “When, where, and how does an event spread over time and space” are questions which many data analyst would like to efficiently exploring from the visual analytics system, thus a VA system should intuitively present and demonstrate those information to the users.

E Event exploration requirements.

E1 Event identification. The system should provide enough information and analysis for user quickly matching the events in social stream with real-life events.

E2 Evolution/Causality analysis. The VA system should provide enough information and analysis for determining what is the triggering events, what is the related events which can happen in the futures. Such information will be valuable for the applications of future forecast systems.

E3 Event diffusion analysis. The system should support on revealing how an event spread over time and space.

E4 Comparative capability. Comparative analysis is strongly demanded in visual analytics. VA system should support the investigation to determine the similarities and differences of events.

E5 Multivariate analysis. To satisfy (V4), the system must be able to analysis the event in multi-perspective levels, such as event, keyword, topic, and sentiment levels.

SocioPedia+ Architecture

This section provides the detailed information of SocioPedia+1 architecture, which is a real-time VA systems for improving event explorations by using social KG. SocioPedia+ was constructed from three different modules operating independently, including social event exploration, spatiotemporal knowledge curation, and visualization, as demonstrated in Fig. 1. Initially, SocioPedia+ collected the data from Twitter based on provided keywords from users. Meanwhile, the location information have been explored by using the user’s location and geo-tag of each collected tweets. Further detailed information will be given in the following sections.

Data presentation

Along with the explosion of information era, an enormous volume of highly diversity data is spreading on the Internet daily. One of the most widely used platforms for exchanging and getting real-time information is social media. We consider each shared message on a social network as a social post. By further analyzing social post properties, plenty of information can be found, such as author, contents of the post, and its publishing time. In brief, a social post can be defined and modeled as follows:

Definition 1 social post

A social post, s, is a part of publication information on a social media. It is modeled by several types of information it contains (1) s=a,c,t,g

where a is the author who published that post, c is content of the post, t is published time and g is geo-tag of the post.

In social networks, the amount of published posts in every second is remarkably enormous, thus creating a social stream of data which has been updated frequently in real-time. However, it is impossible to analyze the whole data stream in a social network because of its huge amount of data. Therefore, we separate the data stream into more specific data stream based on keywords. Each keyword can represent for a data stream, where content of all the posts in that data stream have to include that keyword. A social data stream can be defined as follows:

Definition 2 social data stream

A social data stream (S) is a collection of social posts that are organized according to when they were published (2) S=k,si∣i∈1,∞,k∈si,∀i<j⇒ti≤tj

where k is the streaming keyword, si is a social post, and si is sorted in ascending order of publishing time ti.

We can observe that the changing of data stream because it can reflex the events in the real world. Owing to the ideal and convenient environment created by social networks, people have recently been used to sharing information, stories, and facts about an event in the social network immediately when that event happens in the real world. A social event is modeled as follows:

Definition 3 social event

A social event E is a specific activity that takes place over time T and in a specific location Q (3a) E=P,O,T,Q

(3b) P=si∣i∈1,n

(3c) T=ts:te

where P is a set of social post, O is event’s topic, ts and te are time start and end of event.

Figure 1 Overview of the system SocioPedia, which includes three major parts: social event exploration, event knowledge curation and visualization.

Each event E is defined considering four factors: the collection of posts which mention about the event, the topic of the event, the time period and place that event occurs. When an event happens, the attention from society would be considerably increased. As a natural habit of modern human, users typically try to spread and discuss the event information on the social networks, thus drives to the significantly increment of the posts related to events within a short time period. Such time period is called as a burst. A burst in a social data stream is defined as follows:

Definition 4 social burst

A social burst B is a time period when there are considerably more postings than there are during other times in the data stream (4) B=t1:t2,w

where t1, and t2 is the start and end time of the burst, w is the weight of the burst, which measures the strength of the burst.

As previously discussed, each event E which attracts communities attentions will create a burst B on the timeline. As a consequence, events can be detected by detecting bursts in the social data stream. The burst period is also the time duration of the event. After detecting the event period, we can collect the posts related to that event. But because the content of these posts is unstructured, computer cannot easily understand. We need to convert the information in these posts to a structured format that is friendly to the computer; therefore, we build a knowledge graph from the event by extracting knowledge from these events’ posts. An event knowledge is defined as follows:

Definition 5 event knowledge

An event knowledge K is a triple set of two entities and their relationships that were taken from a social media post (s) during an event (E) (5) K=re1,e2,E

where r is relationship between entities e1, and e2.

Social event exploration

As mentioned in previous sections, to mitigate the issue of huge data volume from social networks, keyword-based data streaming is an efficient solution which have been applied in many reported works. In this work, three following tasks have been conducted to implement the social stream for SocioPedia+:

• Collecting Twitter data from all of countries in the world, in real-time through Application Programming Interface (API)

• Pre-processing raw data to achieve a higher quality data without emojis, redundant words, URL links,

• Storage data to database.

After obtaining a clean data, we perform data classification according to the top trending trends in each country. Two approaches are provided for data classification, including (i) classify following topics, and (ii) classify following current trends. To realize the data classification following the first approach, knowledge extracted from DBPedia has been used to determine related topics. However, the data collected from DBPedia typically have to face with the problem caused by their long release cycles, which could last for several months. Therefore, the freshness and up-to-date level of knowledge collected from open knowledge source might become inaccurate, making users have inaccurate information. To prevent this scenario, SocioPedia+ utilizes the second classification technique. Initially, the collected tweets were analyzed to extract to the most popular terms, based on n-grams evaluation. As such, we can have a briefly overview on which terms are recently an interesting topics on the social stream. In SocioPedia+, the extracted popular terms are considered as the topmost attracted trends and further operations will be implemented following these extracted trends.

After successfully extracting the trends on the social stream, SocioPedia+ performs the next tasks of event detection, which is based on the conventional burst detection technique. Owing to this technique, we can successfully determine the bursty time period. Social posts are then only collected within the bursty time period which allows users to reveal more information about ongoing events.

The burst detection method has been detailed introduced in Kleinberg (2003), which is based on state transformation. The data stream is segmented into a sequence of time points with each time point is a short time period (1 h, 1 day, 1 week). The total number of posts at each time point t is represented as dt. Meanwhile, rt is used to represent how many postings are connected to a certain topic. The ratio of posts that are relevant to a given topic and total posts at a given time is rt/dt.

With each state representing a different time point in the data stream, the burst detection algorithm will fit the sequence of states to the sequence of time points. Each stage depicts the probability of the desired occurrence occurring at the relevant time point. We consider the baseline state and the bursty state to be the only two possible states. The baseline condition denotes a low probability of occurrences, whereas the bursty state denotes a high probability of events. The overall proportion of post the target topic is the baseline state probability: (6) p0=RD

with R represent for the overall number of posts and D represent for all posts in the social data stream: (7a) R= ∑t=1nrt,

(7b) D= ∑t=1ndt

with n is total of time points. The baseline state probability multiplied by the constant s yields the bursty state probability: (8) p1=s×p0.

The purpose of the burst detection technique is to identify state of the social data stream given the sequence of observed proportions. To find the optimal state sequence which fit to time point sequence of the data stream, a cost function c is defined as follows: (9) c= ∑t=1nσi,rt,dt+ ∑t=0n−1τit,it+1

with c is the cost function, i is the state of the sequence. Meanwhile, i = 0 corresponds to the baseline state, i = 1 corresponds to the bursty state, t is the time point, τ is a cost function for assessing the change from one state to the next, and sigma is a function to assess the quality of fitting between the observed proportion and the predicted probability of each condition. The following is a definition of the sigma function: (10) σi,rt,dt=−lndtrtpirt1−pidt−rt.

If the observed proportion fits the anticipated probability of a state, this function gives a low number, and vice versa. The τ function is defined as follows: (11) τit,it+1=it+1−it×γ× lnn,it+1>it0,it+1<it

where gamma is used as a hyper-parameter used to assess the difficulty of transitioning from lower state into higher states. According to this function, moving up a state has a cost, while staying in the same state or moving down a state has no cost. The state transition that minimizes the cost function c is the ideal state transition. We use the Viterbi algorithm (Forney, 1973) to determine the best sequence. The cost function is initialized at time 0 as follows: c0(0) = 0, and c1(0) = ∞. We determine the cost function c for each condition at the following time point t: (12) cit=σi,rt,dt+ minlclt−1+τl,i

where l represents the state at time point t = t − 1 before. By repeating this procedure, the data stream’s final time point t = n will yield the minimal cost function. We use back-tracking after obtaining the minimal cost function to identify the state sequence that corresponds to the minimum cost function. In order to determine the burst period in the data stream, after obtaining the ideal state sequence, we concatenate the neighboring time points that correspond to the bursty state. An event is associated with each burst. The burst’s weight or strength are calculated as follows: (13) w= ∑t=t1t2σ0,rt,dt−σ1,rt,dt

where t1 and t2 are the burst’s start and end time. This weight indicates that, relative to the baseline condition during the burst length, the cost is lower while we are in a bursty state.

The last function in this module is spatiotemporal sentiment analysis. To determine the sentiment of each events and sentiment variations of the users from different countries, we utilize the NewsMTSC (Hamborg & Donnay, 2021) for implementing in SocioPedia+. This library allows determine the sentiment in a sentence with the target as a given input. This means that for a same sentence, we can get different sentiments depending on the input. These data are later stored in the database to serve the next tasks.

Spatio-temporal knowledge curation

Event knowledge extraction

After extracting an event time period, a set of posts which are related to the event from the data stream can be extracted. Nevertheless, collected data in this step is in the textual forms with huge volume of data. As such, it is quite challenge for exploration process with computer. To mitigate this issue, we first extracting the KG from these posts and then filter out the valuable information. Such KG extraction can extract the most important information of the events while keeping all contents in a structured form which is more convenient for solving other tasks.

The collected contents of each post is initially cleaned in the social event exploration step. Each post will be broken up into several sentences. Then, using a variety of techniques, we extract entities and relations from each sentence. The dependency tree, POS tags, and named entities in each sentence will serve as the foundation for the rules. The Algorithm 1 presents the extraction procedure. After obtaining a triple for each sentence that includes subject, object, and relation, we join these triples to create a knowledge graph in which each node represents a subject or an object and each edge represents the relationship between the two. This knowledge graph will show information about an event.

 Input: A sentence S     Result: subj, predicate, obj     subj ←∅; predicate ←∅; obj ←∅;     subjdep ← [nsubj,psubj,csubj];     objdep ← [dobj,iobj,pobj];     tokens ← ListTokenInS;     i ← 0;     while i < length(tokens) do         if tokens[i].head == tokens[i] and (tokens[i].pos == V ERB or     tokens[i].pos == AUX) then        predicate ← tokens[i];    end    if tokens[i] ∈ subjdep and tokens[i].head == predicate then        subj ← tokens[i];    end    if tokens[i] ∈ objdep and tokens[i].head == predicate then        obj ← tokens[i];    end    i ← i + 1;     end                      Algorithm 1: Knowledge extraction.

Temporal factor extraction

In reality, knowledge is temporally variable, and many relationship are just temporary (Liu, Hua & Zhou, 2021). For instance, the link between Donald Trump and the United States only holds true between January 20, 2017, and January 20, 2021. By directly extracting the temporal information (dates, durations, time, etc.) from words, traditional approaches can extract the “temporal factor”. Data from social networks, however, tends to be brief, unorganized, and almost never includes the temporal factor in their contents. To determine the proportion of posts in the data we gathered that had temporal information, we constructed a straightforward model based on Named Entity. The results of Table 2 show, on average, only 23,36% of posts contain temporal information. Therefore, it is not entirely effective to extract temporal information from social networks using standard approaches.

Table 2 Collected tweets dataset description.

Datasets	Collection time	#Tweets	#Time_info	
	From	To			
ESG	Dec.01, 2021	Feb.28, 2022	622.091	113.150	
Ukraine	Feb.28, 2022	March 06, 2022	2.147.789	502.865	
Chelsea	April 20, 2022	April 24, 2022	659.698	149.012	
G7	Feb.28, 2022	April 13, 2022	469.751	134.757	
Korea	June 2, 2022	June 20, 2022	10.706	2.563	

In this study, the time-valid periods of each knowledge are extracted by quantitatively considering occurrence and diffuse-degree features of each knowledge. This indicates that the information will remain accurate as long as it is posted, discussed, and shared on social media. The term-occurrence and difusion-degree are defined as below:

Definition 6 term-occurrence

Assume that the is divided into J time window, the term-occurrence score denoted as TO(j) time period describes the number of triple set K appearance over a time window slot j (14) TOj=twjK

where twjK is the set of tweets contains triple set K in time-slot j.

Definition 7 diffusion-degree

The diffusion degree score denoted as DD(j), describe the diffuse level of the news calculated based on the retweet and share feature of Twitter (15) DDj=rtwjK

where rtwjK is the set of retweets which contains triple set K in time-slot j.

Definition 8 term validation score

The term validation score describes how important a triple set K is in a given time period. Based on this score, we can evaluate a time period is a valid time or not (16) TVj=wTO×twjK+wDD×rtwjK

where wTO, and wDD is weighing factor for term occurrence and diffusion-degree.

The term validation score signal can then be constructed by the set of all scores at each individual time window j (17) TVj≡TVi,TVi+1,TVi+2,…,TVN.

A time period can be evaluated as important time or not by comparing TV(j) at time-period with a threshold.

Knowledge enrichment

Knowledge graphs integration graphs is one of the key objectives for knowledge graph curation. Due to the social knowledge’s extraction from various different events, they are now quite unlinked from one another. As a result, it is challenging to completely comprehend and do additional analysis on such highly unlinked data. For instance, an event explorer would be curious about how an event begins, what knowledge might be the cause of one event, what the information gathered is about, and how the knowledge of one event relates to that of other events. On the other hand, a highly interconnected dataset is extremely important for tasks of reasoning analysis and future forecast system. There have been several reported works on link prediction and relationship search for knowledge graph curation. However, the majority of the works utilized deep learning-based techniques, thus requiring high computing volume. Although the performance is considerably good, these approaches are computation-intensive processes and not suitable for building a real-time and automatic framework. For those reasons, to achieve a highly interconnected knowledge graph in SocioPedia, we mapped all extracted knowledge from the events together by performing a relationship search between entities using the knowledge from DBpedia. The relationship between two entities can be divided into two types: (i) direct relationship and (ii) indirect relationship. The direct relationship denotes a connection between two entities that has an impact on one another. Entity A, for example, has an impact on entity B and vice versa. The direct relationship also depicts a relationship between two entities, however, this time they are derived by a third entity rather than each other directly. For example, entity A affects entity B, and entity B affects entity C. Entities A and C have an indirect relationship, i.e., through entity B.

Visualization Techniques

As aforementioned, the visualization interface of SocioPedia+ has been optimized into two levels for efficiently demonstrating social events and social knowledge in both time and space domains. At initial stage, an overview of tweets distribution following time and countries have been first presented. Based on the collected data, SocioPedia will automatically extract list of trends by analyzing the most popular n-gram and list of suggested topics from DBPedia. From this list, users can choose the trends which they would like to perform more analysis. Graphs of tweets proportions for each trend following time and space are provided for making the users have the most insightful overview. SocioPedia+ automatically detects the events, their time period, and highlight them on the timeline visualization with different colors. In addition, a list of detected events is provided to allow user to analyze all events or one by one. On the other hand, SocioPedia+ analysis and visualization are strongly based on the extracted knowledge from events, thus in this visualization, the number of important knowledge in each event has also be included. Further in-depth analysis and visualizations from two perspective views of social events and social knowledge has also been provided, which will be described in the following sub-sections. More detailed information can be found on the demo2 description of SocioPedia+ and will be presented in the following sections.

Spatiotemporal event exporation

For intuitively visualizing social events characteristics, a VA system should support all required tasks of event explorations and also provide a multi-perspective analysis. Figures 2 and 3 demonstrate the SocioPedia+ visualization optimized for social event explorations. Initially, the event distribution over time is given in Fig. 2A. In this visualization, the detected events are listed in vertical axis whereas the horizontal axis is for timeline. Events are visualized as a line chart with different colors and different lengths which is respective with their time period. By hovering into each events, a list of detailed information can be shown, including event names (represented by the most frequent mentioned knowledge) and event time periods. By displaying the multiple events in a unify timeline, event sequences can be observe to provide an initial overview for the users, who are interested in analyzing the evolution of events. Furthermore, the relation from different events can later be derived by observing and analyzing their extracted knowledge which will be discussed in later sections.

However, although a list of event sequences have already been depicted in Fig. 2A, it is hard for users determining which countries are interested in those events. Certainly, it is almost impossible that an event can attract the attentions from all countries in the world. Therefore, Fig. 2B provides a view point from events distribution both in time and space domains which is observed by the tweets proportions from different countries. In this visualization, the vertical axis is presented for the tweets proportion whereas horizontal axis is for timeline, the tweet proportions of different countries are depicted with different colors, as indicated in figure legend. As such, users can determine which countries are interested in each event and also can quantitatively compare the concern level based on tweet proportions of those countries. Besides that, by zooming out each event time period, spatial event diffusion can be observed by evaluating the bursty time points of different countries in one event. For instances, one event can attract a lot of attention from one country from a time period, after a certain time of spreading information, other countries later start feeling interested about the events and thus the tweet proportions of those countries are sequentially increased and reach a peaks.

It is well-known that sentiment analysis is a crucial task to gain insightful understand about social events. Nevertheless, only few VA systems for event explorations have included sentiment analysis into their visualizations due to the challenge on efficiently integrating the sentiments information into events visualization. Moreover, to gain the most insightful understanding, both temporal and spatial aspects of sentiment analysis should be considered. SocioPedia+ proposed a novel event-centric visualization which have the similar form with a nested pie-chart, as demonstrated in Fig. 3A. Particularly, the event number is designated to center circle whereas the second layer of annular ring refers to the countries which show the interests on that event. The most outer annular ring is constructed from different annular sectors representing for event sentiments from different countries. For instances, Event 1 has attracted attentions from seven different countries whereas most of them feel neutral (not good not bad) about the event. This visualization provides the users the ability to promptly aggregate the spatial-distribution of sentiment, thus providing an in-depth look on what are the attitude of users from different countries on each event. Besides that, the annular length of each sectors from both countries rings and sentiments rings are constructed and calculated to efficiently providing a comparative perspective view on how the events and sentiments distribute in spatial domain. Meanwhile, Fig. 3B provides SocioPedia+ visualization to display the sentiment variations over time and space. Owing to this visualization, users can understand how people’s attitude around the worlds change on an event and also might determine the reason for that variation.

Figure 2 Screenshot of event distributions following time and space of SocioPedia+.

(A) Temporal event distribution. (B) Event distributions following time and space.

Figure 3 Screenshot of sentiment analytics visualization following time and space of SocioPedia+.

(A) Event sentiment analysis following different countries. (B) Sentiment variations of events following time and space.

Multi-perspective event analysis

The ability of multi-perspective analysis on event explorations has recently been limited in VA systems due to the difficulty on analyzing and efficiently visualizing enormous volume of information collected and extracted from social data. Most of reported VA systems only provide the perspective view from keywords and topic analysis whereas other information have been ignored. As a consequence, the lack of multi-perspective information have constrained the ability of VA systems and made them hardly satisfy all required tasks of event explorations. In SocioPedia+, we extend the number of multi-perspective analysis with sentiment and social knowledge analysis both in time and space domains. While the visualization for spatial and temporal sentiment analysis have already been presented in previous sections, this section will discuss more detailed on the visualization for spatiotemporal social knowledge analysis. SocioPedia+ analyze the social stream by calculating the most popular n-gram and considering them as trends. Meanwhile, topics is suggested by deriving information from DBPedia. Both trends and suggested topics are visualized in the overview visualization of SocioPedia+.

Recent reported VA system for event explorations typically provide topics and keyword analysis. Although such analysis can provide some information, that information might not be adequate since the keywords or extracted topics can only provide one part of information. In the comparisons, the knowledge graph (KG), which is represented in form of triples of subject, predicate, object or quadruple of subject, predicate, object, temporal factor, can provide more insightful and valuable information on events owing to the entities and the relations which can be provided by KG. However, efficiently visualizing KG in a real-time VA system is considerably challenging, thus there is no reported VA systems providing KG as one of the multivariate analysis. SocioPedia+ supports the users to efficiently extract important knowledge from collected data and provides different visualizations expressing both time and space information to help users have the most in-depth understanding and analysis on events.

Static visualization. The extracted knowledge from social networks are considerably noisy and highly diffuse due to the unstructured and short-text form of collected contents from social networks. Besides that, considering all social knowledge on the analysis processes might struggle users because they have enormous volume and not all extracted knowledge are important with the events. To handle those issues, SocioPedia+ completes the KG by finding the relationship between unlinked entities to create a highly linked dataset and provides an intuitive visualization to qualitatively evaluate the freshness (old/new), the importance level and connectivity of extracted knowledge. Figure 4 demonstrates SocioPedia+ static visualization for displaying the extracted knowledge from social events. A typical graph visualization constructed from edges and nodes are used to represent each SPO whereas temporal information is placed in each SPO information box. SocioPedia+ uses different colors to represent for different events. Therefore, multiple-color nodes refer to the knowledge which have been mentioned in several events while single-color ones refer to the knowledge which have been only mentioned in one event. Meanwhile, gray color is a special case which is used for representing that knowledge is an old knowledge existed in other open knowledge sources. Dimensions of nodes are different to intuitively emphasize the important level of each knowledge, evaluated based on their mentioned frequencies. Further details including event lists and number of knowledge mentions in each event can be observed by hovering into each respective node and edge. This visualization feature could be meaningful for users for promptly deciding if an extracted social knowledge is worth for further investigation. Besides that, by emphasizing important knowledge of an event, users can use SocioPedia+ for identify important and valuable knowledge to support event identifications, comparison, predictions processes. It should be noted that as aforementioned in “Knowledge Enrichment” section, the extracted SPO from social network show a highly diffuse. Therefore, we proposed a simple knowledge enrichment method for mapping the extracted social knowledge with knowledge from open knowledge source, such as DBPedia. However, in Fig. 4, since the volume of linked knowledge is considerably huge and might cause confusion to readers, we have turned off the algorithm just for a better demonstration.

Timeline visualization. Static visualization has displayed the temporal factor of knowledge in the information box. However, such visualization might constraint user analysis due to the huge data volume of collected knowledge from social networks. Besides that, it is hard to have the most insightful view and comparison between different temporal knowledge while using static visualization. In SocioPedia+, we proposed a social knowledge timeline visualization for addressing those challenges, as demonstrated in Fig. 5. The collected knowledge will be displayed in the vertical left axis with the typical form of a SPO whereas timeline is represented by horizontal axis. SPOs are arranged following the orders of appearance time, in which the SPOs having the earliest start time are listed on the top. The temporal information of each social knowledge is visualized with the similar form of bar charts. By simultaneously displaying the important time period of all knowledge, more thorough analysis can be performed, including investigating and comparing the length of time period as well as observing the mentioned frequency changes over time. Meanwhile, owing to visualizing all collected knowledge following one timeline, SocioPedia+ allows the users to quickly compare the temporal features of different knowledge. Such visualization could bring the benefits for the applications of event evolution analysis, in which the users typically process with the sequence of events. By monitoring the dynamic changes of events over time and identify the knowledge similarity between different events, users can identify which events is the initial one, which events created other events, and which events might happened in the future. As such, this function of SocioPedia+ could bring a considerable advantage for event sequence analysis and event forecast systems. More detailed information can be found on the demo description of SocioPedia+.

Dynamic timeline visualization. The previous section has discussed about the challenge of conventional visualization on intuitively displaying the events characteristic variation and their evolution over time. SocioPedia+ provides a dynamic timeline visualization, in which node-based visualization is used for displaying social knowledge and the node dimensions are represented for how importance that knowledge is, as shown in Fig. 6. Owing to the dynamically animated dimension variations, users can quickly determine the importance of each social knowledge in a certain time period as well as observe its variation. Consequently, this visualization can give users more insightful information to monitoring the knowledge of each events and investigate their variations along with the variation of events.

Evaluation

In this work, we develop a web-based client browser application that uses HTML5, JavaScript, and Django (https://www.django-rest-framework.org/) framework for the back-end development. Plotly (https://plotly.com/) and vis.js (https://visjs.org/) libraries are applied to visualize the chart and knowledge graph in the system. We use PostgreSQL, which is a powerful and open source object-relational database system, as database management system. It provides high performance and scalable functions to manage a massive volume of data.

Figure 4 SocioPedia+ static visualization for collected social knowledge from events.

Figure 5 SocioPedia+ social knowledge timeline visualization.

Figure 6 SocioPedia+ social knowledge dynamic timeline visualization.

Case study

We conducted an evaluation and discussion on a dataset collected from countries around the world during the period from September 17th, 2022 to October 14th, 2022 with the keyword “Korea” as the input. The collected dataset contains 7,356 tweets from 118 countries. From the top trending topics, we choose “ballistic missile” for our case study to verify the usefulness of SocioPedia+ for exploring social events including event identification, event sequence visualization, diffusing analysis, comparative capability, and causality analysis (E1–E5).

There are eight events detected in this time period, as can be observed from the spatial event distribution visualization in Fig. 2A (E2). Particularly, SocioPedia detected “Event 1”, which was last from September 24th to September 25th 2022 and attracted the attention from users coming from seven different countries or regions, including United States, Pakistan, Iceland, Nigeria, United Kingdom, Japan, and Taiwan (Province of China). The most frequently mentioned knowledge in “Event 1” was {North Korea, fired, ballistic missile}, which has a highly representative value can allow the users to promptly catch the event (E1). By matching this extracted knowledge with event time period, a ground truth event of “North Korea fires ballistic missile ahead of US VP Harris visit” can be easily extracted from website of Reuters, a highly reputation journal. Meanwhile, spatial event distribution can be observed in Fig. 3A to reveal the diffusion of events (E3). As can be observed, while “Event 1” attracted attentions from seven countries, the number of attracted countries were reduced over time. At time period of “Event 2” and “Event 3”, the attracted countries were six and one, respectively. However, at “Event 4” time period, the number of interested countries was significantly increased to 13 countries. This could mean that a significant event might happened in this time period, thus robustly boosted up the event diffusion.

To have a more in-depth analysis, the multivariate analysis provided by SocioPedia+ could be considerably meaningful to compare different events (E4, E5). Because the number of attracted countries and the number of extracted knowledge were simultaneously reduced at time period of “Event 2” and “Event 3”, we can conclude that the ground truth event happening in these time periods are unimportant and might be a sub-event caused by “Event 1”. Meanwhile, it is worth to further analysis “Event 1” and “Event 4” with the support of SocioPedia+ multivariate analysis. Firstly, the most frequently mentioned knowledge of “Event 4” was {North Korea, fires, ballistic missile over Japan}, which can be matched with ground truth event of “North Korea fires ballistic missile over Japan”, as reported in website of BBC news. Interestingly, although “Event 1” and “Event 4” shared a quite similar social knowledge, the number of attracted countries of “Event 4” are approximately twice of the number from “Event 1”. In addition, the sentiment analysis have shown that the users from more countries expressed negative responses with “Event 4” than “Event 1”. From these observation, the users can have a brief conclusion that “Event 4” and Event 1“ have similar feature on ”creating conflicts“ but the seriousness of ”Event 4“ is considerably higher than ”Event 1”, thus attracts more attention and more negative reactions. It should be noted that although the causality have not been demonstrated in particular, SocioPedia+ can also provide a highly-linked social knowledge dataset with multivariate analysis, thus can bring considerable benefits for the applications of forecast system.

User evaluations

To evaluate SocioPedia+, we performed a laboratory survey3 to gather user opinions and feedbacks. We invited 16 participants, including two Post-Doctoral, one PhD students, 10 master students, and three users working for IT companies; all users are not co-authors of this article and have background on computer science. The main purpose of this investigation is to evaluate the task accuracy and the effectiveness of SocioPedia+ on the applications of event explorations. The experiments are divided in two sections for validating the task accuracies and for evaluating the effectiveness/usefulness of system. At the beginning, we let the participant learn a short tutorial on how to use SocioPedia+ and then let them exploring some events by themselves. After they finished the tasks, several question which is respects to each event exploration tasks were given to the attendant to verify if the user can explore and derive the correct information from the system. These data was later processed and aggregated to derive the accuracies of each tasks performed by the participants. Meanwhile, the satisfaction and evaluation on system usefulness and effectiveness are scored with a rating system. The user study tasks are based on basic tasks of event explorations, listed as following

T1 Event identification

T2 Event evolution/causality analysis

T3 Diffusing analysis

T4 Comparative analysis

T5 Multivariate analysis

On the other hands, for evaluating the system usefulness and visualization effectiveness, we have asked the participants to rate the system on the scale of 1 to 5, based on several key criteria as follows:

C1 Overall satisfaction

C2 Easy-to-use

C3 Evaluation on the intuitiveness of social events visualization

C4 Evaluation on the intuitiveness of sentiment visualization

C5 Evaluation on the intuitiveness of social knowledge visualization

C6 Evaluation on event analysis

C7 Evaluation on events comparison

C8 Evaluation on multivariate analysis

C9 Evaluation on multivariate comparison

In general, the task accuracies and participant feedbacks are both quite positive, as demonstrated in Fig. 7A. Particularly, the investigated results have shown that users complete task T1, T2, and T3 with noticeable high accuracies of 82.64%, 100%, and 95.31%, respectively. Meanwhile, although the accuracies of the user experiments on task T4 and T5 is slightly lower, the figures are sequentially are 81.25% and 74.22%, which is still considerably good. For T5, we have discussed with all participants to determine the reason which made users perform a lower accuracy on this task. After investigation, we identified that the number of multi-perspective analysis have made the users a little confused and some of the participants cannot identify full perspective view presented in the SocioPedia+ system. This can be easily understand because all users have just learned how to use the system for only 5–10 min. After we explained each visualization stage again, the participants have completed task T5 with perfect accuracies. Figure 7B presents the results of user evaluation for SocioPedia+ based on the criteria of C1–C9. In general, participants has highly ranked the proposed system in most of criteria. Particularly, all users are extremely attracted by the visualization intuitiveness on different levels of event, knowledge, and sentiment (C3–C5). Meanwhile, all participants found the multi-perspective analysis and comparison of SocioPedia+ are extremely useful. Among all criteria, C2 have shown the lowest point of 4.0/5.0. After thoroughly discussed with participants, the lower rating of C2 is caused due to the confusing which the system have made to the participants while investigating task T5.

Figure 7 User evaluation for SocioPedia+ system.

(A) Summary of task accuracy. (B) Users evaluation results.

Besides the user evaluations, the participants have been interviewed to gain more detailed opinions. As numerically expressed from the results of evaluations, the detailed opinions are generally good and most of the users found SocioPedia+ as a useful visual analytics system for event exploration. However, they have also raised some opinions about it would be better if SocioPedia+ allows users to present a geographic map for spatial distribution visualization whereas some other users suggested SocioPedia+ should include the arrow into the visualization of sentiment analysis to better reveal the causality of events. These comments can be a very good idea for the future improvement of SocioPedia+.

Conclusions and Future Work

In this article, we presented SocioPedia+, a novel visual analytics system for improving spatiotemporal event exploration. By introducing one more dimension of social knowledge graph into the system multivariate analysis, the process of event explorations can be significantly enhanced and thus enabling system capability on performing full required tasks of visual analytics and social event explorations. To provide the most comprehensive overview and most detailed analysis, SocioPedia+ provides an intuitive visualization with multi-perspective spatiotemporal analysis from social event level to knowledge level. The evaluation results including case study, user evaluation, and detailed interview with participants have strongly demonstrated the usefulness and visualization effectiveness of SocioPedia+. However, there remains several limitations on the proposed SocioPedia+ system. It can be observed that although the extracted SPO show a highly representative value and easy to understand, the algorithm we used for extracting SPO have some limitations which it hardly identifies between same words using uppercase/lowercase letter or same words but using different tenses. This is a limitation causing to a slightly redundancy in extracted knowledge which we have to consider for improving in the future works. In addition, user interactive is quite limit in this version of system. More analysis interactive function should be introduced into the system to enhance the experience and analysis of the users. In the future work, we plan to upgrade the SocioPedia+ for initially improving the users interactive and perform more detailed analysis to reveal the more information related to events. Particularly, although SocioPedia+ provides a multivariate analysis which can be extremely meaningful for analyzing event causality analysis, the visualization for this task have been still limited in this design. As such, SocioPedia+ can extend its applications and might bring the benefits to a broader range of users.

Additional Information and Declarations

Competing Interests

Author Contributions

Data Availability

1 SocioPedia+ System, http://recsys.cau.ac.kr:8086/sociopedia/

2 Sociopedia demonstration, t.ly/yrgk

3 SocioPedia+ survey, https://forms.gle/6cjTnr6A7LgqDp4EA

The authors declare there are no competing interests.

Tra My Nguyen conceived and designed the experiments, performed the experiments, analyzed the data, performed the computation work, prepared figures and/or tables, authored or reviewed drafts of the article, and approved the final draft.

Hong-Woo Chun performed the computation work, prepared figures and/or tables, authored or reviewed drafts of the article, funding, and approved the final draft.

Myunggwon Hwang conceived and designed the experiments, analyzed the data, prepared figures and/or tables, authored or reviewed drafts of the article, and approved the final draft.

Lee-Nam Kwon performed the experiments, analyzed the data, authored or reviewed drafts of the article, and approved the final draft.

Jae-Min Lee conceived and designed the experiments, prepared figures and/or tables, authored or reviewed drafts of the article, and approved the final draft.

Kanghee Park conceived and designed the experiments, analyzed the data, authored or reviewed drafts of the article, and approved the final draft.

Jason J. Jung conceived and designed the experiments, performed the experiments, analyzed the data, performed the computation work, prepared figures and/or tables, authored or reviewed drafts of the article, funding, and approved the final draft.

The following information was supplied regarding data availability:

The data and system is available at GitHub: https://github.com/kecau/sociopedia.

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
