# Peer review of "SocioPedia+: a visual analytics system for social knowledge graph-based event exploration"

_PeerJ Computer Science, doi:10.7717/peerj-cs.1277_

## Round 0.1 · original submission · Major Revisions

The authors should address the review comments to revise and resubmit the manuscript.

Reviewer 1 ·

Basic reporting

The language is ok, but I have found some stylistic and language mistakes in the text. I strongly advice the language to be checked and mistakes corrected by Nativr speaker or professional transtion expert before publishing the paper in the journal.
The literature review is good and all sufficient information to understand the importance of the topic has been provided by authors.
I also think that the graphical side of the paper should be improved. The text is very small and the high compression factor (poor quality) in figures makes it hard to read even if the reader enlarges the pdf file. Also I not sure that the Figure 2 is necessary since all its parts are presented in subsequent figures.

Experimental design

The topic of the paper is described in sufficient details to understand its imprtance for social media data exploration and analysis. The temporal aspect of the data is important and have to be included in all approaches to making a synthesis of information provided by various social media.

Validity of the findings

The novelty of the proposed approach and its importance is clearly stated in the paper. I think that some comparison with existing approaches which can be used by online users should be presented in the paper. For example, there should be some discussion of the user's functionality of the proposed solution with respect to the existing ones - a comparison done by users in a similar manner that the SocioPedia's evaluation was provided in the paper.
I have also some detailed questions to authors concerning the proposed solution.
1) The data exploration and information synthesis from the social media is an interesting issue. However, the value of information may be questionable if fake news are also included. Can your spatio-temporal method deal with the issue of fake news? Can you detect its source and spread in the social networks? Can you comment how can the authors deal with this problem?
2) Is the Burst-time alone enough to compute the true timeline of events? If something new happens, it becomes the most interesting issue for social media users, but decrease of interest does not automatically suggests that the event truly ended. This concerns important and long-term events like recent COVID epidemics and the War in Ukraine. The media interests of these events is less than months ago, but they are still important.

Additional comments

I have some additional questions concerning the results presented by authors.
1) In the graph representation of knowledge generated by your approach, there are many disconnected nodes (SPOs) and some apparent redundancy. For example the event "Japanese citizens take shelter" is listed twice - the reader is not sure if it is a redundancy which was not reduced the the system or two distinct events that happened at different times. There are other examples in the figure.
Also the example, the authors mentioned - "N. Korea fires missile over Japan" is confusing. Does the system classified the same event reported by different sources as two events or does it concern two different events (at different times)?

Reviewer 2 ·

Basic reporting

Very well written however its suggested to add more recent and relevant references.

Experimental design

Is it possible to further elaborate on the experiment specially considering the reproducibility of your work.

Validity of the findings

comparative analysis with recent state of the art is recommended

Additional comments

The paper has promising results but the authors are suggested to consider the comments to further enhance the quality of the paper.
-abstract needs to mention clearly the contributions
-introduction should have each sections details and how the reader will be able to grasp the basics
-the literature review is not comprehensive enough please ensure to have a table for comparison and also to include a discussion to formulate your problem
-Experimental analysis may be detailed so it is reproduceable
-limitations and highlights of the contributions may be given
-kindly elaborate on how you have validated the study and how does it compare with the state of the art
-finally all figures may be captioned appropriately and details may be given to enhance the quality of tables and figures for readibility.

---

## Round 0.2 · accepted · Accept

The reviewer has accepted the submission.

Reviewer 2 ·

Basic reporting

The authors have improved the paper.

Experimental design

The authors have improved the paper.

Validity of the findings

The authors have improved the paper.

Additional comments

The authors have improved the paper.